# Impact of ATM and DNA-PK Inhibition on Gene Expression and Individual Response of Human Lymphocytes to Mixed Beams of Alpha Particles and X-Rays

**DOI:** 10.3390/cancers11122013

**Published:** 2019-12-13

**Authors:** Lei Cheng, Beata Brzozowska-Wardecka, Halina Lisowska, Andrzej Wojcik, Lovisa Lundholm

**Affiliations:** 1Centre for Radiation Protection Research, Department of Molecular Biosciences, The Wenner-Gren Institute, Stockholm University, 106 91 Stockholm, Sweden; 2Biomedical Physics Division, Faculty of Physics, University of Warsaw, 02-093 Warsaw, Poland; 3Institute of Biology, Department of Radiobiology and Immunology, Jan Kochanowski University, 25-406 Kielce, Poland

**Keywords:** DNA damage response, radiation, gene expression

## Abstract

Accumulating evidence suggests a synergistic effect in cells simultaneously exposed to different types of clustered and dispersed DNA damage. We aimed to analyse the effect of mixed beams of alpha particles and X-rays (1:1 dose of each) on DNA damage response genes in human peripheral blood lymphocytes isolated from four donors. Two donors were compared upon inhibition of ATM or DNA-PK and at different sampling times. qPCR was used to measure mRNA levels of *FDXR*, *GADD45A*, *BBC3*, *MDM2*, *CDKN1A*, and *XPC* 24 h following exposure. Generally, alpha particles and mixed beams were stronger inducers of gene expression compared to X-rays, displaying saturated versus linear dose–response curves, respectively. Three out of four donors responded synergistically to mixed beams. When two donors were sampled again one year later, the former additive effect in one donor was now synergistic and no significant difference in intrinsic radiosensitivity was displayed, as determined by gamma-radiation-induced micronuclei. ATM, but not DNA-PK inhibition, reduced the radiation-induced gene expression, but differently for alpha radiation between the two donors. In conclusion, synergy was present for all donors, but the results suggest individual variability in the response to mixed beams, most likely due to lifestyle changes.

## 1. Introduction

Understanding the response of cells to DNA damage is highly relevant in the context of carcinogenesis as well as of cancer therapy where DNA-damaging therapies are commonly used. The most critical types of DNA damage—DNA double-strand breaks—display distinct spatial patterns and complexity depending on the ionising radiation (IR) quality [1]. Sparsely ionising radiation with low linear energy transfer (LET) like X-rays or gamma rays mainly interact with DNA via reactive oxygen species produced from water radiolysis, predominantly giving rise to simple, dispersed lesions. Alpha particles (helium nuclei) and heavy ions primarily induce direct damage along short and dense linear tracks resulting in lesions of high complexity [1]. Clustered or complex damage is characterised by a slower repair compared to low LET-induced, simple damage [2]. IR-induced clustered damage comprises two or more lesions (i.e., double-strand breaks (DSBs) as well as single-strand breaks or base damage on both strands) within less than 20 bp of DNA [3]. In addition, the repair of these types of damage may cause new DSBs or other forms of misrepair, resulting in increased genomic instability [4].

Two main DNA repair pathways handle DSB repair, primarily the canonical/classical non-homologous end-joining (cNHEJ), active in all cell cycle phases; and homologous recombination (HR), which requires a sister chromatid as template and is available in S/G2-phase during replication, for end-resected DSBs as well as for transcription-associated DSB repair [5,6]. cNHEJ is controlled by DNA-dependent protein kinase (DNA-PK), facilitating the multistep process initiated by Ku70/80 binding to broken DNA ends and terminated by ligation by the DNA ligase IV complex [7]. The rare autosomal recessive disorder ataxia telangiectasia (A-T), caused by biallelic mutations in the ataxia-telangiectasia mutated (*ATM*) gene, is characterised by an extreme radiosensitivity, cancer predisposition and neurodegeneration [8]. The gene product ATM protein kinase is regarded as a master regulator of DNA damage response (DDR) with many hundreds of downstream substrates having roles in cell-cycle arrest, apoptosis and DNA repair, via both the cNHEJ and HR pathways [9]. In case the cNHEJ is not active, DSB repair can take place via microhomology-mediated end joining (MMEJ), also called alternative NHEJ (alt-NHEJ) [7]. Similarly, when HR is non-functional, single-strand annealing (SSA) may take place [6]. Resection is required for both the SSA and MMEJ processes, leading to deletions [6].

An interesting question is, how do cells react when clustered and dispersed damage is induced simultaneously? Scenarios of mixed IR exposure occur in numerous cases. In the environment, radiation comes from the ground or from building materials in the form of gamma radiation (uranium, thorium disintegration products) and alpha radiation (inhaled as radon gas) [10]. During airplane and space flights, humans are exposed to cosmic radiation which is a blend of several different qualities, mainly protons, alpha particles, and electrons, which then interact in the atmosphere and decay to muons, electrons, photons, neutrons, and protons [10]. In proton therapy and intensity-modulated radiotherapy (IMRT), secondary neutrons may produce recoil (secondary) protons which have a higher LET than the primary protons, generating a mixed beam together with the low-LET protons [11] or IMRT photons [12].

The complexity of exposure to multiple stressors has recently been highlighted in diverse fields [13]. Not only DNA-damaging agents of different types interact, so also do chemical agents as determined by the so-called cocktail effect. Indeed, people are often exposed to a combination of several DNA-damaging or toxic agents, unlike the single-agent exposure scenarios that are commonly tested in laboratory settings. This was also the case after the Hiroshima–Nagasaki bombings, where people were exposed to a mix of high-energy neutrons and gamma rays [14]. Nuclear weapon tests and accidents such as Chernobyl and Fukushima Daiichi lead to chronic exposure, mainly from dispersed ^131^I and ^137^Cs [15,16]. The chronic exposure to gamma radiation was mixed with exposure to chemical toxins and UV radiation from the sun.

A number of radiation-responsive genes, mainly involved in the p53-signaling pathway, have previously been identified [17]. We have used a selection of those in two earlier studies to assay radiation response [18,19]: BCL2 binding component 3 (*BBC3*), cyclin dependent kinase inhibitor 1A (*CDKN1A*), ferredoxin reductase (*FDXR*), growth arrest and DNA damage inducible alpha (*GADD45A*), MDM2 proto-oncogene (*MDM2*) and XPC complex subunit, DNA damage recognition and repair factor (*XPC*). These genes are generally induced rapidly (at 4 h post irradiation), peaking at 24 h and starting to decline at 48 h, as determined by our report on the effect of mixed beams on human peripheral blood lymphocyte (PBL) gene expression [18]. The gene induction was consistently highest following 2 Gy mixed beam exposure in the tested donor (three independent experiments) at all three time points [18].

In this study, we used a panel of radiation-induced genes to assay the response to mixed beams in human PBL from four donors. We used inhibitors towards ATM and DNA-PK to advance our mechanistic understanding, and evaluated the inter-and intraindividual response to mixed beams in two of the donors. The study is a continuation of our analyses of cellular effects of simultaneous exposure to alpha radiation and X-rays [18,20,21,22,23].

## 2. Results

### 2.1. Dose Response of Radiation-Induced mRNAs after Exposure to X-rays, Alpha Particles and Mixed Beams

In order to verify the effect of mixed beams, relative mRNA levels of *BBC3*, *CDKN1A*, *FDXR*, *GADD45A*, *MDM2* and *XPC* were measured at 24 h post irradiation with 0 to 2 Gy, in PBL from four donors individually. The dose–response curves of relative mRNA levels for six genes in Donors 1–4 are plotted in Figure 1 and Figure 2A. Figure 2B shows the summarized results of average values for all four donors. Generally, all six genes in the four donors’ PBL showed a positive relationship with the increased doses. Alpha particles and mixed beams induced similar mRNA levels for most of the genes, and were much higher than X-rays, especially at lower doses. The dose–response curves of alpha particles and mixed beams were fitted to the same equation, while the dose–response curve of X-rays was much more linear. In Donor 1, Donor 2 and the summarized result, the dose–response curves of alpha particles and mixed beams almost overlap with each other (Figure 1A,B and Figure 2B). In Donor 3, mixed beams induced an even higher gene expression than alpha particles, while in Donor 4, the result was the opposite (Figure 1C and Figure 2A). This indicates that the mixed exposure interaction might display individual differences.

Both alpha particles and mixed beams induced a strong increase in gene expression at lower doses, and the increase halted at higher doses and turned into a saturated curve, which led to larger differences between mRNA levels induced by alpha particles (or mixed beams) and X-rays at lower doses compared to higher. This difference was significant, not for all cases but for certain genes at several doses in certain donors. Among all genes, the expression of *FDXR* was most sensitive to IR. The relative mRNA level of *FDXR* was induced at the highest fold change by IR among all six genes, and the levels for alpha particles and mixed beams were significantly higher than for X-rays at 0.5 Gy in all four donors and the summarized result, as well as at 1 Gy in three of four donors and 2 Gy in two of the four donors. *GADD45A*, *MDM2* and *XPC* were also significantly more strongly induced by alpha particles and mixed beams compared to X-rays at 0.5 Gy and 1 Gy (except *GADD45A*) in the sum of all donors, while *BBC3* displayed higher levels in two donors. *CDKN1A* was less influenced by IR compared to the other genes, and the relative mRNA level (fold change) was around 2, even at the highest dose of 2 Gy. There were no significant differences in *CDKN1A* expression among the three types of IR at any dose.

### 2.2. Mixed Beam Effect on Individuals Analysed Using Envelopes of Additivity

The *FDXR* gene was chosen for the envelopes of additivity analysis due to its high sensitivity to radiation exposure, as shown by us and others [24]. It was used to verify if the mixed beam exposure induced a synergistic, additive or sub-additive effect. The method is described in the Materials and Methods section. The envelopes of additivity were constructed for doses inducing a low, medium or high level of excess. For Donors 1 and 3, mixed beams showed a synergistic effect at all three levels in inducing the *FDXR* gene expression (Figure 3, panels A, B, C and panels G, H, I). The synergistic interaction was strongest at a low level of gene expression fold and weakest at a high level. For Donor 2, mixed beams induced a higher than expected gene expression level at low and medium excess levels (panels D and E) which indicated a synergistic effect, but had an additive effect at a high excess level (panel F). For Donor 4, mixed beams induced an additive effect at low level (panel J), but sub-additive effect at medium and high levels (panel K and L). Again, the results suggest that individual differences may be present after exposure to mixed beams.

### 2.3. Mixed Beam Effect Changes in the Same Donor at Different Sampling Times

We chose to perform further studies using ATM or DNA-PK inhibition in two donors only, based on the differences seen between Donor 3 (as a representative of the synergy) and Donor 4 (representing additivity). The gene expression level after exposure to 1 Gy X-rays, alpha particles and mixed beams was checked in the PBL of Donor 3 and Donor 4, both in the dose–response experiment (series 1) and ATM or DNA-PK inhibition experiment described below (series 2). The interval time of the two experiments was about 1 year. The results are summarized and compared in Figure 4. For Donor 3, the relative mRNA levels were stable. Only *BBC3* (mixed beams) and *MDM2* (alpha particles and mixed beams) expression levels were significantly increased, and *CDKN1A* levels were more variable in series 2. The alpha particles and mixed beams induced the highest increase in mRNA level and X-rays induced the lowest increase in mRNA level in both series 1 and series 2, which means that the synergistic effect of mixed beams was kept at two different timepoints (panel A). For Donor 4, the mRNA levels of *FDXR* significantly increased after exposure to all three IR types in series 2 compared to series 1 (panel B), while *GADD45A* expression levels significantly increased after exposure to mixed beams, and *MDM2* mRNA levels increased after exposure to alpha particles and mixed beams when comparing series 1 and series 2. Alpha particles induced the highest relative mRNA level in series 1, but in series 2, the mixed beams’ effect increased and induced a similar mRNA level as alpha particles, which was generally much higher than X-rays. The effect was most prominent for *MDM2*. The result indicated that the mixed beam effect transferred from additivity in series 1 to synergism in series 2 in Donor 4.

### 2.4. Micronucleus (MN) Assay in Donor 3 and Donor 4

An MN assay was applied here to analyse if Donor 3 and Donor 4 were different in their intrinsic radiosensitivity. The dose–response curves of the two donors for gamma-radiation-induced micronuclei are compared in Figure 5. Donor 4 had a slightly higher MN level at higher doses compared to Donor 3, but the difference was statistically non-significant. The results in Figure 4 and Figure 5 suggest that factors other than the individual genotype matter for the radiation response.

### 2.5. Gene Expression Modulation Using ATM or DNA-PK Inhibitor

A dose of 1 Gy X-rays, alpha particles and mixed beams was used to examine if inhibition of ATM (inhibitor KU55933) or DNA-PK (inhibitor NU7026) influenced gene expression in Donor 3 and Donor 4 24 h after exposure (Figure 6 and Appendix A). Inhibition of ATM tended to reduce DDR gene expression induction after exposure, but the inhibition of DNA-PK did not. mRNA levels of *FDXR* and *GADD45A* were significantly lower when ATM was inhibited compared to without inhibition after X-ray exposure, while the expression level of *MDM2* was significantly lower after inhibition of *ATM* and mixed beam exposure in Donor 3 (Appendix A). In Donor 4, mRNA levels of *FDXR* and *GADD45A* were significantly lower upon ATM inhibition compared to without inhibition after X-ray or alpha radiation exposure. Gene expression was unchanged by the addition of inhibitors only (without IR), as displayed by relating data to average controls without inhibitor (Appendix A). In Figure 6, the relative level reflects each gene’s mRNA expression during inhibition of ATM or DNA-PK, normalised to its mRNA expression without inhibitor treatment after the same exposure. In Donor 3’s PBL, inhibition of ATM decreased relative levels of *CDKN1A*, *FDXR* and *GADD45A* after exposure to X-rays and mixed beams, while the relative level of *CDKN1A* was also significantly lower after alpha radiation. Interestingly, the relative level of *FDXR* showed a significant increase by treatment with DNA-PK inhibitor (NU7026) (panel A). In Donor 4’s PBL, the decrease in relative level by inhibition of ATM was clearer than in Donor 3, as all genes were lower on average when using ATM inhibitor KU55933 (panel B) together with all radiation qualities, with the exception of BBC3 after exposure to alpha particles and X-rays. Significant decreases were seen for all genes after exposure to X-rays in Donor 4. The inhibition of DNA-PK had no influence on the total gene expression of the six genes in Donor 4. The total gene relative level is summarized in panel C and panel D for Donors 3 and 4, respectively. The lower DDR gene expression induction upon inhibition of ATM was significant after exposure to X-rays and mixed beams in Donor 3, while it was significant for all three exposure types in Donor 4. The decrease tended to be larger after exposure to X-rays compared with alpha particles and mixed beams, but it was not significantly different. Inhibition of DNA-PK had no significant influence on gene expression induced by IR in both donors’ PBL.

## 3. Discussion

In this study, we made a thorough effort to characterise the cellular response to mixed DNA damage in the form of ionising radiation, which is an inducer of well-characterised and time-controlled DNA damage. We assayed four individuals, where blood was taken and processed separately at three time points during two time periods. The repeated sampling for series 2 (at three time points again), where we could use the controls for the ATM and DNA-PK inhibition as additional samples, allowed for the analysis of intra-individual differences.

Overall, the first set of experiments displayed clear synergy for most donors, also when the sum of the four donors was analysed. This result is in agreement with our earlier results on synergy in human PBL using micronuclei [22], chromosomal aberrations [23], comet assay and radiation-induced gene expression [18] as endpoints. Synergy was also detected at the level of 53BP1 foci and the activation of DNA repair proteins using human U2OS osteosarcoma cells [20] and as the hindered mobility of foci in mixed-beam-exposed cells as demonstrated by live cell imaging [21]. However, in certain cases, we observed the additivity of both radiation types: at the level of clonogenic cell survival in AA8 Chinese hamster ovary cells [25] and at the level of gamma-H2AX foci in human VH10 cells [26]. Several studies from other laboratories have also analysed the effect of mixed low-and high-LET radiation beams, some detecting synergism and other additivity (summarised in [13]), yet the reason for this variance is unclear.

When each donor was plotted separately, a clear difference in pattern between Donor 3 and Donor 4 was observed, which led us to perform repeated analyses of PBL collected from those donors only as examples of different response characteristics. Interestingly, while the response for Donor 3 remained unchanged at the second sampling time, the results for Donor 4 did change, revealing synergy at the second sampling time. The fold increase was also generally closer to that of Donors 1–3 (ex. 20-fold vs. ca. 10-fold increase in *FDXR* for Donor 4), and this was not due to differences in basal levels (control real-time PCR raw Cp values were similar between Donors 3 and 4). In order to test if the individual differences seen between the two donors was related to differences in their intrinsic radiosensitivity, dose–response curves for gamma-radiation-induced micronuclei were compared. No differences were seen, which may suggest a lack of a genetic component in the different responses.

The genetic contribution to variation in individual differences in sensitivity to ionising radiation is not known. Although rare monogenic diseases exist that are responsible for consistently elevated sensitivity to radiation [27], individual response to radiation among healthy individuals is regarded as a complex trait [28]. Complex traits are influenced by the genetic background, the environment and stochastic variations in molecular interactions [29]. Vral et al. have shown that intra-individual variability in the frequency of radiation-induced micronuclei can exceed the inter-individual variability [30]. Also, a lack of correlation was reported in radiosensitivity of PBL and fibroblasts correlated from the same individuals [31], suggesting that the impact of environmental and stochastic components dominates over the genetic component of individual response to radiation. In turn, which one of the two plays a more important role? Bearing in mind that the present study was based on four individuals of which repeated blood sampling was only carried out for two individuals, the results point towards environmental factors. The reason for this is that a variable response was seen only in one donor and only between samplings separated by a time period of ca. 1 year. Results from blood samplings carried out within one series (i.e., during a time period of a few weeks) were consistent. Needless to say, this conclusion is speculative because of the low number of analysed individuals.

Radiation-induced gene expression in PBL is regarded as a promising retrospective biological dosimeter [32,33]. Due to its high responsiveness to radiation, the *FDXR* gene appears as a particularly suitable biodosimeter in cases where sample collection is possible within a short time after exposure [24]. Our results support this conclusion, as the *FDXR* gene showed the highest level of induction in PBL of all donors. What is interesting is the difference in shape of dose–response curves between X-rays, alpha particles and mixed beams. While the former was generally linear, it saturated for both alpha particles and mixed beams. From the perspective of biological dosimetry, it points towards the necessity of generating separate calibration curves for different radiation types. It also appears that the sensitivity to detect increments in alpha and mixed beam doses above ca 0.5–1 Gy is limited.

In this study, we wanted to specifically inhibit the two main kinases involved in IR-induced DSB repair, ATM and DNA-PK, to further evaluate the mechanistic aspects of the response to mixed beam radiation. ATM is needed for the slow component of the DNA repair process in G1 phase involving end resection, which accounts for ca. 15% of the X-ray repair, but a larger proportion for high-LET damage [6]. In addition, ATM has been shown to be required for the repair of DSB in heterochromatin, via signals that enhance relaxation of the tightly packed chromatin [34]. We could clearly see a lower general gene induction in cells where ATM was inhibited, compared to those without inhibitor. Interestingly, this was the case in Donor 4 for most genes, irrespective of radiation quality, while for Donor 3 there was no clear distinction with or without inhibitor after alpha particle exposure. The effect of mixed beams was relatively close to the X-ray effect and very similar between the donors when comparing all genes (Figure 6C,D). These results instead indicate that ATM is an even stronger factor when 100% contribution comes from high-LET damage such as for alpha particle radiation, which is consistent with previous suggestions of an important role of ATM in the response to complex, high-LET-induced damage [35]. A 50% contribution as in the mixed beam scenario does not seem to provide any intermediate response between X-rays and alpha in this respect. At this point we cannot determine the cause of this difference in response upon ATM inhibition between donors. Since the intrinsic response to low-LET damage was similar between the donors based on MN data, we would expect the same for high-LET damage (without ATM inhibition) as well. This would however require further study on larger groups of individuals, including repeated sampling at separate time periods. There are also processes mediated by ATM which could possibly influence the cellular homeostasis without being related to DNA repair; examples are insulin signalling and responses to hyperthermia, hypoxia and hypotonic stress [36]. Considering the important role of ATM, an even higher reduction in IR-induced gene activation than the average ca. 50% reduction upon ATM inhibition could perhaps have been expected, based on studies using p21^CIP1/WAF1^ protein levels as endpoint in other cell systems [37,38]. No other studies using ATM or DNA-PK inhibition were found assaying DDR gene expression at a similar time point; perhaps the high radiosensitivity of lymphocytes merely overloads the potential of the ATM inhibitor. This is supported by results from lymphocytes where numbers of γH2AX, phospho-ATM and phospho-DNA-PK foci were reduced by ca. 50–70% upon ATM inhibition [39].

There was no obvious reduction in response when using a DNA-PK inhibitor for any of the radiation qualities; rather, there was a slight increase for some genes. There are reports on a hyperactivation of ATM and an amplification of p53 response when DNA-PK is inhibited, which possibly could contribute to the observed phenomena [9]. The use of another DNA-PK inhibitor (NU7441) in MCF7 breast cancer cells did not modulate the p21^CIP1/WAF1^ or MDM2 protein levels at 24 h after IR, which is consistent with the unchanged pattern for most of our genes [38]. Further analysis of the combined use of these inhibitors would be helpful, yet dual ATM and DNA-PK inhibition acted similarly to single ATM inhibition (KU55933) and reduced the IR-induced activation of the p21^CIP1/WAF1^ and MDM2 proteins [38].

## 4. Materials and Methods

### 4.1. Blood Collection and Irradiation for Gene Expression Analysis

Blood collection and irradiation methods were modified from earlier work [18]. Fresh peripheral blood was collected by venepuncture in heparin blood collection tubes (VACUTEST Kima, Arzergrande, Italy) from four healthy male non-smokers, just before experiments (we have an ethical permit with the number 2010/27-31/1 from the Ethical Committee at the Karolinska University Hospital). Whole blood was diluted 1:1 in RPMI 1640 medium (Sigma-Aldrich, R5886, Stockholm, Sweden) and 250 μL was exposed or sham-exposed on round polyamide (PA) discs (155 mm in diameter, custom-constructed in the Institute for Energy-JRC, Petten, Netherlands) as described earlier [22], covered with a 2.5-μm-thick Mylar foil lid and spread all over the disk in an even layer.

Blood samples were collected from each donor on three occasions within a period of ca. 2 weeks for dose response experiments. In Donors 3 and 4, the sampling was repeated after a break of ca. 1 year in order to carry out experiments with ATM and DNA-PK inhibitors. The two sampling series are referred to as series 1 and series 2 to visualise individual response variability.

An irradiation facility consisting of an alpha irradiator AIF 08 (^241^Am, 50.0 ± 7.5 MBq) on the top and an YXLON SMART 200 X-ray tube (operating at 190 kV, 4.0 mA, no filtering) underneath, which allows exposure with alpha particles and X-rays simultaneously as described by Staaf et al. [25], was applied for each exposure. A movable shelf in the mixed beams facility was used to position the cells on the PA disc at a defined distance from the alpha source to control the exposure on and off. The dose rate of the alpha source was 0.223 Gy/min by calculation at the entrance to the cell suspension and the average LET was 90.9 ± 8.5 keV µm^−1^. The dose rate of X-ray tube was 0.068 Gy/min in the bottom and 0.052 Gy/min in the top position of the movable shelf. Doses of 0, 0.5, 1 and 2 Gy were applied for the dose–response study of alpha particles, X-rays and mixed exposure. A dose of 1 Gy was used for the detection of the influence of ATM or DNA-PK inhibitors on gene expression. Mixed beam exposure contained half dose each of alpha particles and X-rays and always started in the top position with simultaneous alpha particles and X-rays exposure. After reaching the intended alpha dose, the shelf was moved down to the bottom position with X-rays on, as the dose rate of alpha particles was higher than X-rays. X-rays were then switched off after reaching the planned X-ray dose [25].

After exposure, blood was collected directly from the disc to 14 mL cell culture tubes (Greiner bio-one, Frickenhausen, Germany), adding 5 times the blood volume of RPMI 1640 medium supplemented with 20% foetal bovine serum (Gibco, Invitrogen, Stockholm, Sweden), 100 U/mL penicillin and 100 μg/mL streptomycin (Sigma-Aldrich, P4333, Stockholm, Sweden), and incubated at 37 °C, 5% CO_2_ for 24 h.

### 4.2. Gene Expression Analysis by qPCR

RNA was extracted using the E.Z.N.A. Total RNA Kit I (Omega Bio-tek, Norcross, GA, USA from whole blood treated by Red Blood Cell Lysis Buffer (Roche, Mannheim, Germany) after irradiation of X-rays, alpha particles and mixed beams following 24 h incubation. cDNA was synthesised from 150 ng RNA using the High-Capacity cDNA Reverse Transcription Kit (Thermo Fisher Scientific Baltics UAB, Vilnius, Lithuania) with random hexamer primers. Primers, cDNA and 5× HOT FIREPol® EvaGreen® qPCR Supermix (Solis BioDyne, Tartu, Estonia) were mixed and real-time PCR reactions were performed in duplicate on a LightCycler^®^ 480, starting at 95 °C for 12 min, followed by 45 cycles of 95 °C for 15 s, 60 °C for 20 s and 72 °C for 20 s. No template control reactions were used to identify PCR contamination. The 2^−ΔΔCt^ method was used for calculation of relative mRNA expression, and melting curve analysis was used for testing primer specificity. The same primers were used as previously [18]: *BBC3* for: TACGAGCGGCGGAGACAAGA, *BBC3* rev: GCAGGAGTCCCATGATGAGATTGTAC; *FDXR* for: TGGATGTGCCAGGCCTCTAC, *FDXR* rev: TGAGGAAGCTGTCAGTCATGGTT; *GADD45A* for: ACTGCGTGCTGGTGACGAAT, *GADD45A* rev: GTTGACTTAAGGCAGGATCCTTCCA; *XPC* for: GCTTGGAGAAGTACCCTACAAGATGGT, *XPC* rev: GGCTTTCCGAGCACGGTTAGA; *MDM2* for: TATCAGGCAGGGGAGAGTGATACA, *MDM2* rev: CCAACATCTGTTGCAATGTGATGGAA; *CDKN1A* for: CCTGGAGACTCTCAGGGTCGAAA, *CDKN1A* rev: GCGTTTGGAGTGGTAGAAATCTGTCA. For *18S*, sequences are given in [40].

For gene expression analysis after treatment of inhibitors, whole blood was incubated with 10 μM ATM inhibitor KU55933 (Sigma-Aldrich) or 10 μM DNA-PK inhibitor NU7026 (Sigma-Aldrich) at 37 °C for 1 h before exposure with 1 Gy alpha particles, X-rays and mixed beams, respectively. After exposure, the incubation was continued with same concentration of inhibitors for 24 h before RNA extraction.

### 4.3. Irradiation and Micronucleus Analysis

The micronucleus (MN) analysis method was performed similarly as previously [41]. Blood samples were collected from two out of four donors who donated blood in gene expression analysis and irradiated by gamma radiation from a ^137^Cs source (0.79 Gy/min, Gammacell 40 Exactor, MDS Nordion, Ontario, Canada) at 0, 0.5, 1, 1.5 and 2 Gy at 37 °C. Whole blood (0.5 mL) was mixed with 4.5 mL complete medium composed of RPMI 1640 medium supplemented with 20% fetal bovine serum, 100 U/mL penicillin and 100 μg/mL streptomycin and phytohemagglutinin (PHA, M, Gibco). The whole blood was cultured at 37 °C, 5% CO_2_. Cytochalasin B (Sigma-Aldrich) was added to each tube after 44 h of incubation at a final concentration of 5.6 μg/mL to inhibit cytokinesis, and the samples were harvested after 72 h incubation. Control samples were also prepared in the same way.

Subsequently, diluted blood was centrifuged and resuspended in warm (37 °C) 0.14 M KCl (Sigma–Aldrich). After incubation at room temperature for 5 min, the cells were centrifuged and fixed in fixative I (methanol:0.9% NaCl:acetic acid (Sigma–Aldrich); 12:13:3) and subsequently in fixative II (methanol:acetic acid; 4:1). Two or three washes with fixative II were performed until the supernatant was clear. Then, the cells were dropped onto clean, dry microscope slides (VWR, Spånga, Sweden), dried and stained with 5% Giemsa (Merck, Darmstadt, Germany).

The numbers of MN were scored manually and blindly using a 40× objective on a Zeiss microscope. MNs were scored in around 500 binucleated cells (BNCs) per slide, and as there were two slides per treatment, a total of 1000 BNCs per treatment were scored.

### 4.4. Statistical Analysis

Relative mRNA levels were used to draw dose–response curves of X-rays, alpha particles and mixed beams. The best fits to the dose–response and repair kinetics curves were identified using GraphPad Prism v5.00 (GraphPad Software, San Diego, CA, USA).

The best fit function to the X-ray data was: *y* = *c* + *ax* + *bx*^2^, where *y* is the relative mRNA level, *x* is the dose and *a*, *b*, *c* are the fit coefficients.

The best fit function to the alpha and mixed beam data was: *y* = *c* + *a*(1−e^−*bx*^), where *y* is the relative mRNA level, *x* is the dose and *a*, *b*, *c* are the fit coefficients.

An envelope of additivity model was applied to analyse the significance of interaction between alpha particles and X-rays in influencing gene expression [42]. This model was constructed using iso-effect plots (isobolograms) based on isoadditivity or heteroadditivity of the effect induced by a combined exposure, when at least one of dose response curves is not linear [43]. If an observed effect is inside of the envelope, it means that the measured data indicate an additive effect. If the measured data point lies below the envelope region, it is a synergistic effect. A data point positioned above the envelope upper edge indicates a sub-additive effect. Here the variable used to describe an effect was excess levels, defined as relative mRNA levels minus 1. The dose–response curve shape which the envelope analysis depends on does not change, but the values on the *y* axis become proportionally scaled.

Differences between relative mRNA levels among irradiation types at different doses were tested by one-way ANOVA. The differences in relative mRNA levels with inhibitors compared to without were tested by the unpaired, two-tailed Student’s *t*-test. The differences in MN frequencies induced by the same dose of gamma rays between two donors were also tested by the unpaired, two-tailed Student’s *t*-test.

## 5. Conclusions

In conclusion, we demonstrated synergy after exposure to mixed beams from alpha and X-ray radiation in four donors, with more saturated dose–response curves for alpha and mixed-beam-induced gene expression. The lack of difference in intrinsic radiation sensitivity towards gamma radiation and difference between sampling periods suggest environmental factors as contributors to the variance between donors. ATM inhibition displayed general but high-LET-radiation dependent differences in radiation-induced gene expression.

## Figures and Tables

**Figure 1 cancers-11-02013-f001:**
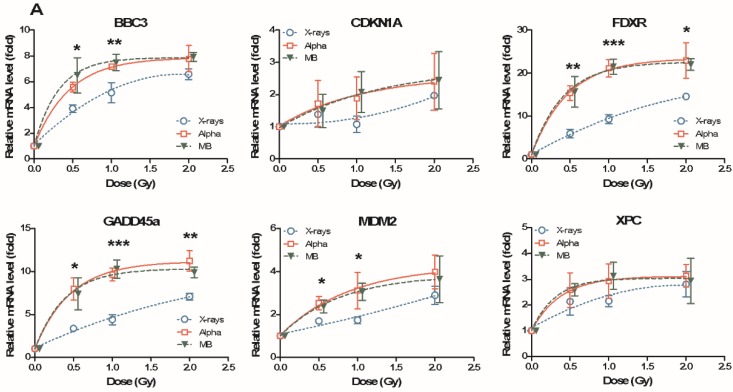
Dose response of DNA-damage-responsive gene expression levels in human PBL after 24 h incubation following exposure to X-rays, alpha particles, and mixed beams in Donor 1 (**A**), Donor 2 (**B**) and Donor 3 (**C**). Relative mRNA level: mRNA expression fold after exposure compared to control. Symbols represent mean results from three independent experiments. Error bars represent standard deviations. Asterisks represent significant differences at the level of * <0.05, ** <0.01, and ***< 0.001 for mixed beams and alpha radiation compared to X-rays (one way ANOVA). Symbols are nudged for transparency.

**Figure 2 cancers-11-02013-f002:**
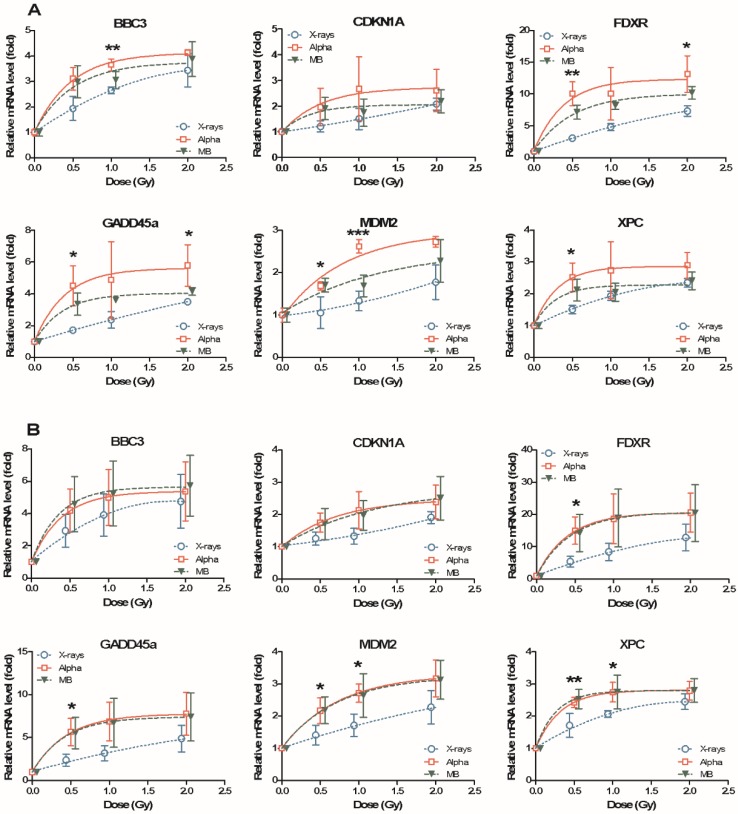
Dose response of DNA-damage-responsive gene expression levels in human PBL after 24 h incubation following exposure with X-rays, alpha particles and mixed beams in Donor 4 (**A**) or as the average of four donors (**B**). Relative mRNA level: mRNA expression fold after exposure compared to control. Symbols represent mean results from three independent experiments (**A**) or from four different donors (**B**). Error bars represent standard deviations. Asterisks represent significant differences at the level of * < 0.05 and ** < 0.01 for mixed beams and alpha radiation compared to X-rays (one-way ANOVA). Significant difference *** < 0.001 to alpha particles (one way ANOVA) at 1 Gy in *MDM2*. Symbols are nudged for transparency.

**Figure 3 cancers-11-02013-f003:**
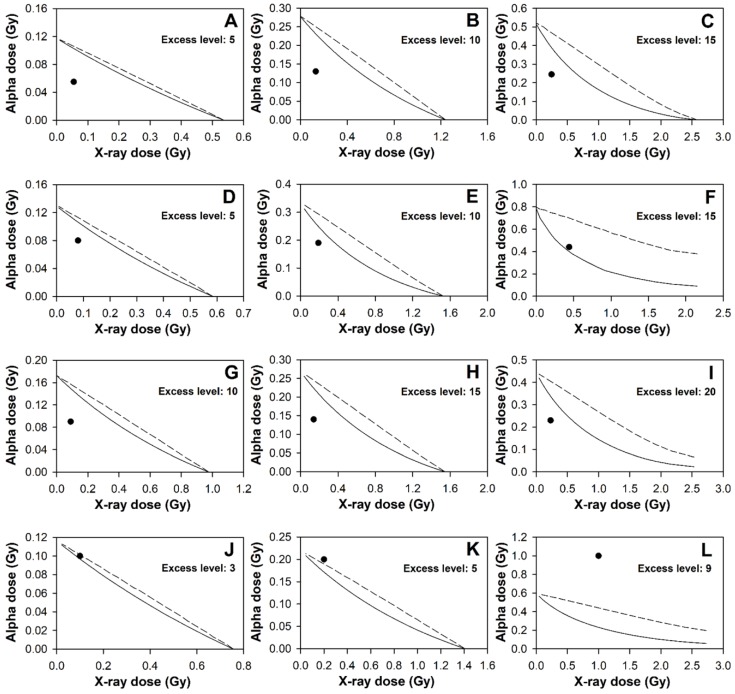
Envelopes of additivity calculated for the expression of the *FDXR* gene in PBL for three chosen excess levels for Donor 1 (**A**–**C**); Donor 2 (**D**–**F**); Donor 3 (**G**–**I**) and Donor 4 (**J**–**L**). Excess levels are fold increase levels minus 1. Excess levels for X-rays were fitted to the equation y = c + ax + bx^2^ and for alpha particles and mixed beams to the equation y = a(1 − e − bx) where x = dose in Gy.

**Figure 4 cancers-11-02013-f004:**
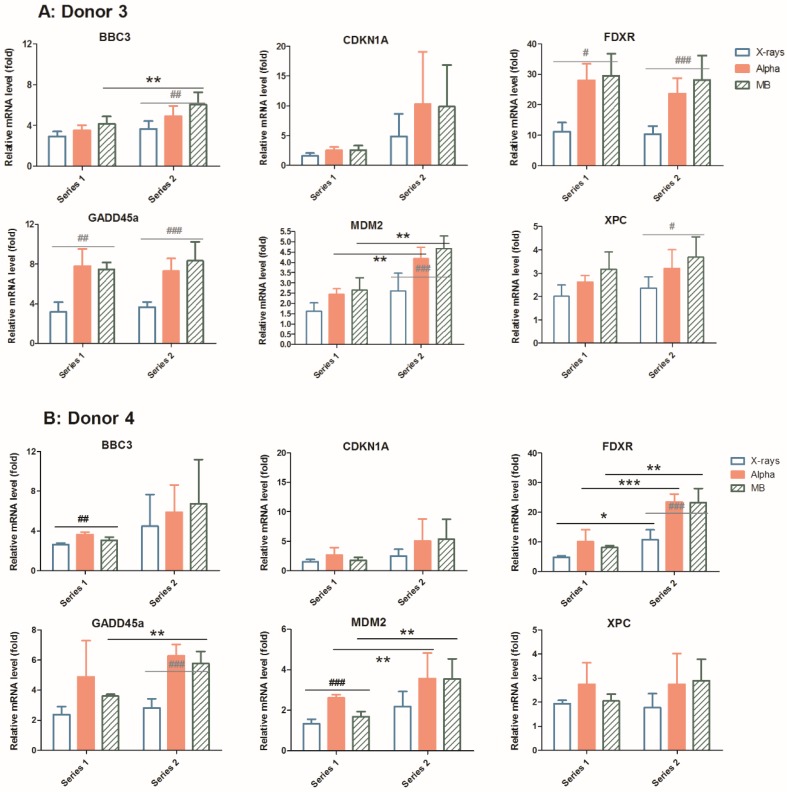
Relative mRNA levels at time periods separated by ca. 12 months (series 1 and series 2), 24 h after exposure to 1 Gy X-rays, alpha particles or mixed beams, respectively in Donors 3 (**A**) and 4 (**B**). Bars represent mean results from three independent experiments. Error bars represent standard deviations. Asterisks represent significant differences at the level of * <0.05, ** <0.01 and *** <0.001 of mRNA levels in PBL 24 h after the same exposure with inhibitor compared to without inhibitor using student t-test. Hashtags represent significant differences at the level of # <0.05, ## <0.01 and ### <0.001 compared within series between different exposures (one-way ANOVA).

**Figure 5 cancers-11-02013-f005:**
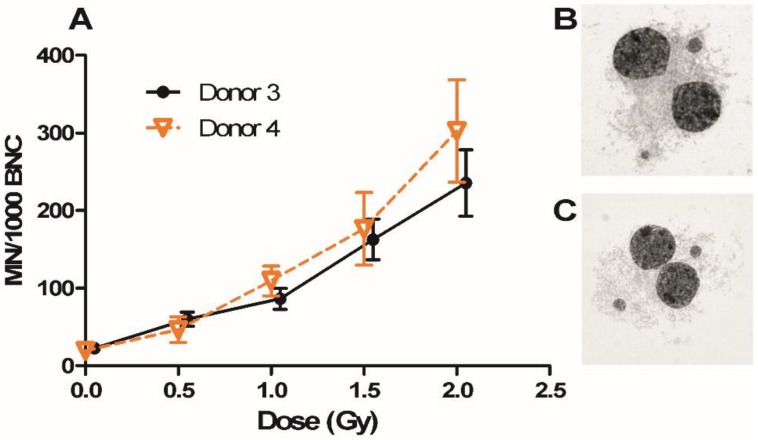
Dose–response curves of micronucleus (MN) frequency in Donor 3 and Donor 4 after exposure to gamma rays (**A**). MN frequencies per 1000 binucleated cells (BNCs) in PBL did not significantly differ between the two donors at any dose. Symbols are nudged for transparency. Error bars represent standard deviations from three independent experiments. Example MN images of 2 Gy samples from Donor 3 (**B**) and 4 (**C**) are presented.

**Figure 6 cancers-11-02013-f006:**
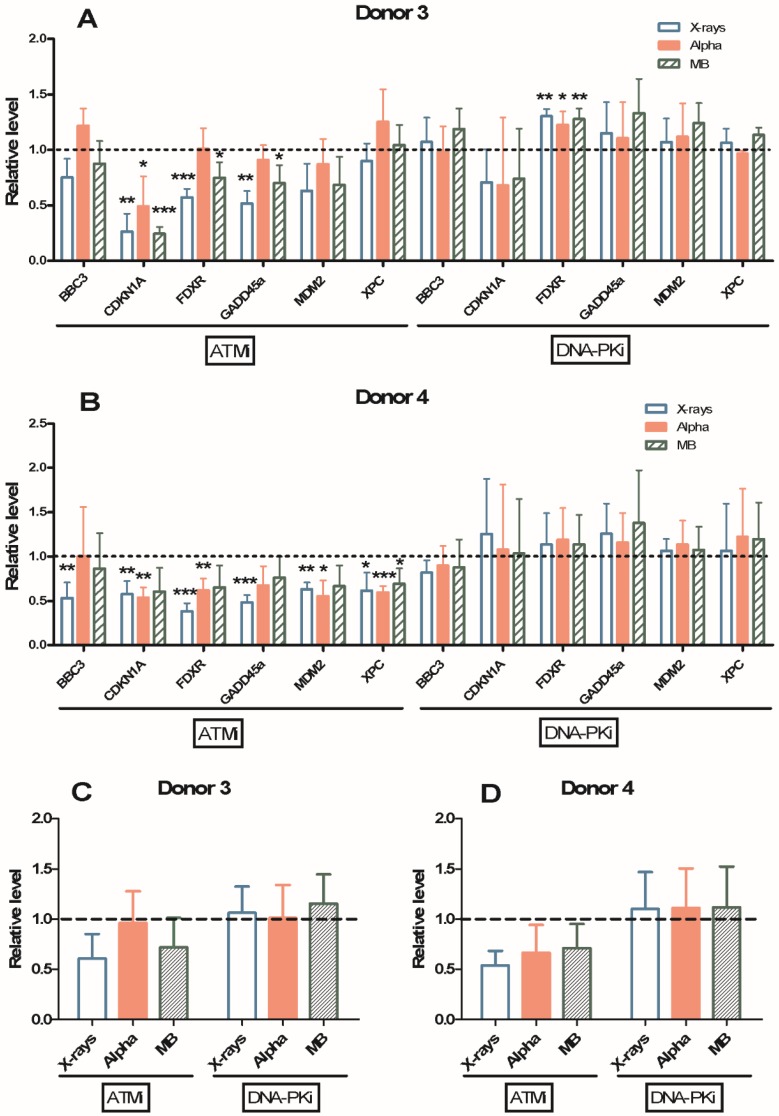
Relative levels of gene expression 24 h after exposure to 1 Gy X-rays, alpha particles or mixed beams, respectively, with or without inhibition of ATM (KU55933) or DNA-PK (NU7026) in Donors 3 and 4. The relative level is the relative mRNA level with inhibitor normalized to that without inhibitor (calculated for each experiment, then average ± standard deviation is presented), for all genes individually for Donor 3 (**A**) and 4 (**B**), or all genes combined when all relative levels for each radiation quality are averaged for Donor 3 (**C**) and 4 (**D**).

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
