# Peer review of "Impact of ATM and DNA-PK Inhibition on Gene Expression and Individual Response of Human Lymphocytes to Mixed Beams of Alpha Particles and X-Rays"

_cancers, 2019, doi:10.3390/cancers11122013_

Round 1

Reviewer 1 Report

The authors measured the effect of different types of radiation on gene expression levels in blood cells and find a synergistic effect of mixed beams of alpha and Xrays in blood from 4 donors. The study is interesting, however, the sample set of 4 donors is limited to support some of the general conclusions and inter/intra sample variations complicate discussion. It is to be seen whether results are generally applicable or whether it is limited to these samples. Furthermore, I have some comments:

Major comments:

-There is a large summary of experiments previously conducted by the same research group in the introduction. This should be shortened and can briefly be included in the discussion when putting it in context also comparing it with results from other research groups.

-The manuscript has many figures showing the same data in a different way. I recommend to merge some of the figures and move additional data to the supplemental section. This will increase readability of the manuscript.

-It is not clearly explained why the second part of the research is only conducted in donor 3 and 4 and not 1 and 2, please specify.

-Figure 10: Explain what can be seen in the different panels.

-It is not clear what the difference is between the data in figure 9 and 10. Can these be merged?

-The first paragraph of the discussion could better be moved to the introduction to put the research into perspective.

-There is repetition of the results in the discussion, please remove.

-The authors draw conclusions on a limited set of samples (4 or only 2). For example, that environmental factors contribute to intra individual changes (line 298). It should be noted that only a limited sample number was examined and this might severely influence the outcome. Also it is suggested that gene expression in PBL can be used as biodosimeter. This research only investigated expression patterns relatively early after radiation. Is it therefore reasonable that this can be used as biodosimeter since you would expect that most samples will be taken at later timepoints.

-ATM is involved in many more cellular processes than DNA repair. This should also be noted by the authors. It might influence the gene expression pattern in a way unrelated to DNA repair.

-The discussion about the DNA-PK inhibitor effects is highly speculative, please provide more evidence or discuss in more detail.

-There are quite a number of grammatical mistakes in the text. Some are listed below, but please read the text carefully and correct.

Minor comments:

-Line 47: requires instead of require

-Page 2: Mixed uses of cNHEJ and NHEJ.

-Line 59: not clear to which ‘latter processes’ is referred.

-Line 68: have instead of has

-Line 78: what end points? Please specify.

-Line 179: results instead of result

-Figure 8: The curve of donor 3 seems to be shifted slightly compared to donor 4. Did this donor indeed receive different doses?

-DNA-PK/ATM inhibitor and the name of the compound are used mixed in the text. Please use either DNA-PK inhibitor or NU (same for ATMi). Also specify in Figure 9 which inhibitor is which as was done in Figure 10

Author Response

Thank you very much for your review of this paper and your helpful comments, our responses are included below (italic):

Major comments:

-There is a large summary of experiments previously conducted by the same research group in the introduction. This should be shortened and can briefly be included in the discussion when putting it in context also comparing it with results from other research groups.

This part has been moved to the discussion. A short reference to previous publications is made in the last sentence of the introduction.

-The manuscript has many figures showing the same data in a different way. I recommend to merge some of the figures and move additional data to the supplemental section. This will increase readability of the manuscript.

Some of the first figures are now merged and fig 9 was moved to supplemental data.

-It is not clearly explained why the second part of the research is only conducted in donor 3 and 4 and not 1 and 2, please specify.

A sentence describing the reasons was added as a start of section 2.3. A similar description was available already in the discussion. Additionally, the decision to focus on two donors was also made due to availability of the same donors at this later time point as well as the number of experimental samples to handle.

-Figure 10: Explain what can be seen in the different panels.

-It is not clear what the difference is between the data in figure 9 and 10. Can these be merged?

Figure 9 and 10 are based on the same data, but in Figure 9 they are presented as relative to control and the gene induction by IR is visible, with or without inhibitor. In Figure 10 they are instead only related to the sample without inhibitor for each radiation type, in our mind more clearly displaying the effect of ATM and DNA-PK inhibition. We have now moved Figure 9 to supplemental data, since we think that Figure 10 is more informative in showing the effects of the inhibitors. Former figure 10 legend has now been extended to better explain the comparisons made.

-The first paragraph of the discussion could better be moved to the introduction to put the research into perspective.

Agreed, this section has now been moved to the introduction.

-There is repetition of the results in the discussion, please remove.

Removed and/or rephrased to link it better to the discussion.

-The authors draw conclusions on a limited set of samples (4 or only 2). For example, that environmental factors contribute to intra individual changes (line 298). It should be noted that only a limited sample number was examined and this might severely influence the outcome.

We are stating that this is based on very few individuals and that it is speculative. If the response would be merely stochastic, the values within those few weeks (three samplings) would be similar as to when sampling was done one year apart. This does not seem to be the case, at least for several genes, therefore we chose to phrase it as “point towards environmental factors”. We have rephrased the sentence about a genetic component to “which may suggest” in the third section of the discussion.

“Bearing in mind that the present study was based on four individuals of which repeated blood sampling was only carried out for two individuals, the results point towards environmental factors. The reason for this is that a variable response was seen only in one donor and only between samplings separated by a time period of ca 1 year. Results from blood samplings carried out within one series (i.e. during a time period of a few weeks) were consistent. Needless to say, this conclusion is speculative because of the low number of analysed individuals.”

Also it is suggested that gene expression in PBL can be used as biodosimeter. This research only investigated expression patterns relatively early after radiation. Is it therefore reasonable that this can be used as biodosimeter since you would expect that most samples will be taken at later timepoints.

This is correct, we have added now an updated ending to this sentence:

“Due to its high responsiveness to radiation, the FDXR gene appears as a particularly suitable biodosimeter in those cases when sample collection is possible within a short time after exposure”

-ATM is involved in many more cellular processes than DNA repair. This should also be noted by the authors. It might influence the gene expression pattern in a way unrelated to DNA repair.

A sentence describing this has been added to the discussion.

-The discussion about the DNA-PK inhibitor effects is highly speculative, please provide more evidence or discuss in more detail.

Two sentences discussing DNA-PK and dual inhibition of both kinases are included in the discussion.

-There are quite a number of grammatical mistakes in the text. Some are listed below, but please read the text carefully and correct.

We have now gone through the text again and made corrections

Minor comments:

-Line 47: requires instead of require

-Page 2: Mixed uses of cNHEJ and NHEJ.

-Line 59: not clear to which ‘latter processes’ is referred.

-Line 68: have instead of has

-Line 78: what end points? Please specify.

-Line 179: results instead of result

Corrected. For line 78, the specific end points where additivity was seen was referred to in the last half of the sentence. The sentence is now rephrased and moved to the discussion.

-Figure 8: The curve of donor 3 seems to be shifted slightly compared to donor 4. Did this donor indeed receive different doses?

No, they received the same doses. The slight shifting of the curve is due to nudging of data points, which is done only to increase the visibility of specific data points and error bars for individual samples. This was explained in the legends for Figure 1-5, but it is now included in this figure legend as well, we are sorry that it was missed:

“Symbols are nudged for transparency.”

-DNA-PK/ATM inhibitor and the name of the compound are used mixed in the text. Please use either DNA-PK inhibitor or NU (same for ATMi). Also specify in Figure 9 which inhibitor is which as was done in Figure 10.

Thank you for pointing this out, we are now only using the terms DNA-PK/ATM inhibitor or DNA-PKi/ATMi within the figures and throughout the text. The compound names are added as well in some places.

Reviewer 2 Report

This study reports the effects of X-rays, alpha particles and mixed beam exposure (1:1) on the induction of the expression of a set of 6 p53-dependent genes in PBLs purified from 4 individual donors. The results presented clearly show inter-individual differences in the response to the different types of exposure. For 2 of the donors, this response evolves in time, as different results were obtained with PBLs sampled one year apart. The authors attribute these differences to lifestyle factors, as they estimate that the findings of similar levels of micro-nuclei induction in both donors precludes the effects of genetic differences. They then use inhibitors of ATM and DNA-PK to analyze the requirement for these signaling pathways in gene induction post exposure. Only the ATM inhibitor was found to have some effects, which differed with exposures.

Overall, this study present very interesting data on the difference in transcriptional response of PBLs to exposure to X-rays, alpha particles and mixed beams, and clearly shows that different patterns of response can be identified in different donors, even though some of them can exhibit the same radio-sensitivity by the micro-nuclei assay. I think however that it is  difficult to attribute these differences only to lifestyle factors on the basis of these results.

I think that the following points should be addressed:

The effects of ATM and DNA-PK inhibition are evident only for a subset of genes, and their effects is not complete, even though the amplified genes are all “involved in the p53-signaling pathway”, and therefore also dependent on ATM activation. Consequently, I am surprised that the ATM inhibitor did not have a stronger effects on more genes after X-rays exposure. Did the authors check that these inhibitors are active in their experimental conditions? Or are such “partial” effects (in terms of target genes and level of inhibition) documented in similar experimental conditions?

I am puzzled by the representation of data in fig 10. Has the normalization been performed on the average gene expression value shown in fig 9. And if yes, how was taken into consideration the very large SD observed for some of the “no inhibitor” points (CDKN1A for donor 3, BBC3, CDKN1A, MDM2 for donor 4) to calculate significance of differences? Or is it that the normalized fold increase is calculated for each of the experiments performed in triplicate, and then the average of the 3 experiments +/- SD is represented? Furthermore, what is represented in panel C and D? please explain in details what this “pattern” is and was established?

Page 6 line 148 : « both alpha particles and mixed beams induced a rapid increase of gene expression at lower doses and the speed slowed down at higher doses… » Gene expression was measured only at one time point post exposure (24h). How then can the author describe a kinetics of gene expression?

Figure 7: is the data for XPC expression in donor 3, series 1 correct? Alpha radiation and MB seem to give the same level of expression (around 3) but in figure 4, the fold increase for alpha radiation is clearly lower.

Page 10, line 238: “in donor 4, the decrease in relative level…as all genes were significantly lower when using ATM inhibitor (panel B)  together with all exposures except BBC3 after alpha particles and X-rays” this statement should be corrected; In Fig 10B, no differences (vs crts?) is seen for several combinations of gene and exposure.

Author Response

Thank you very much for your review of this paper and your helpful comments, our responses are included below (italic):

I think that the following points should be addressed:

The effects of ATM and DNA-PK inhibition are evident only for a subset of genes, and their effects is not complete, even though the amplified genes are all “involved in the p53-signaling pathway”, and therefore also dependent on ATM activation. Consequently, I am surprised that the ATM inhibitor did not have a stronger effects on more genes after X-rays exposure. Did the authors check that these inhibitors are active in their experimental conditions? Or are such “partial” effects (in terms of target genes and level of inhibition) documented in similar experimental conditions?

This is an interesting question, and clearly a stronger effect of ATM inhibition could have been expected. We did not check the activity of ATM (phospho-ATM) in these samples, the inhibitors were however used according to an established setup (1 h pretreatment, then kept during the post IR incubation time) and concentration (10 µM), as previously described (Cmielova J et al Int J Rad Biol 2013, Lundholm L et al CDDis 2013). The only published mRNA data on DDR genes we were able to find upon ATM/DNA-PK inhibition did display an almost complete abolishment of (transient) transcriptional induction of GADD45a, but no change for CDKN1A at 1 h post IR in human HF1 fibroblasts. Those cells were pre-treated with KU55933 for 15 min and then irradiated with 2 Gy and kept for up to 4 h after irradiation in the presence of KU55933 (Narayanan et al. Scientific Reports, 7, 43598, 2017). We suggest that this may be due to the high radiation sensitivity of lymphocytes, a few sentences about this were included in the discussion.

Some data is available on protein levels, but the Westerns were not always quantified. ATM inhibition in dental pulp mesenchymal stem cells reduced the protein levels of the CDKN1A gene product p21 by ca 80% after 2 Gy at 24 hours post irradiation (Cmielova J, et al. Int J Radiat Biol 2013). p21 protein levels were also clearly lower upon KU55933 pretreatment at 24 h post radiation, and a similar pattern was seen upon KU55933 combined with another DNA-PK inhibitor in MCF7 cells (Ćmielová J, et al. Med Oncol 2015). This last example is also included in the discussion.

I am puzzled by the representation of data in fig 10. Has the normalization been performed on the average gene expression value shown in fig 9. And if yes, how was taken into consideration the very large SD observed for some of the “no inhibitor” points (CDKN1A for donor 3, BBC3, CDKN1A, MDM2 for donor 4) to calculate significance of differences? Or is it that the normalized fold increase is calculated for each of the experiments performed in triplicate, and then the average of the 3 experiments +/- SD is represented? Furthermore, what is represented in panel C and D? please explain in details what this “pattern” is and was established?

For the former figure 10, the figure legend has now been updated. The values +inhibitor were first related to –inhibitor for each experiment and radiation quality, averages+/-stdev are presented in A and B. Since a high IR-induced (vs control) induction with inhibitor usually correlated with a higher IR-induced induction without inhibitor for a particular experiment, the standard deviations became smaller in former Fig 10 than 9. For C and D, the relative RNA levels with vs without inhibitor for each radiation quality and gene were all averaged (18 data points in total per radiation quality and inhibitor.

Page 6 line 148 : « both alpha particles and mixed beams induced a rapid increase of gene expression at lower doses and the speed slowed down at higher doses… » Gene expression was measured only at one time point post exposure (24h). How then can the author describe a kinetics of gene expression?

Thank you for pointing this out, this is of course not concerning kinetics and has been corrected: “Both alpha particles and mixed beams induced a strong increase in gene expression at lower doses and the increase hailed at higher doses and turned into a saturated curve”

Figure 7: is the data for XPC expression in donor 3, series 1 correct? Alpha radiation and MB seem to give the same level of expression (around 3) but in figure 4, the fold increase for alpha radiation is clearly lower.

We are very sorry for this mistake, thank you for spotting this. The correct values were in figure 4 and the former figure 7 has now been updated.

Page 10, line 238: “in donor 4, the decrease in relative level…as all genes were significantly lower when using ATM inhibitor (panel B)  together with all exposures except BBC3 after alpha particles and X-rays” this statement should be corrected; In Fig 10B, no differences (vs crts?) is seen for several combinations of gene and exposure.

Corrected, and clarified that significant changes for all genes are only present after X-rays.

Reviewer 3 Report

I have some concerns before it can be published in the manuscript. Authors should address these points before resubmitting.

Concerns:

qPCR was used to measure mRNA levels of FDXR, GADD45a, BBC3, MDM2, CDKN1A and XPC 24 hours following exposure. It would be beneficial to perform the western blot and check the levels with specific antibodies, which will make the results more conclusive. However, I liked the dose-response curve of DDR genes.
2. Micronucleus assay figure is needed other than the graph. It will be nice to see the micronuclei. The methodology of this assay is very well written, but one figure will be good to put in the main figure.
3. The authors should add an experiment to include both of the inhibitors (DNA-PK mixed with ATMi) for the DDR gene response to irradiation. This would tell whether hyperactivation of ATM upon DNA-PKcs inhibition in response to DNA damage occurs in donors 3 and 4 or not.

Author Response

Thank you very much for your review of this paper and your helpful comments, our responses are included below (italic):

I have some concerns before it can be published in the manuscript. Authors should address these points before resubmitting.

Concerns:

qPCR was used to measure mRNA levels of FDXR, GADD45a, BBC3, MDM2, CDKN1A and XPC 24 hours following exposure. It would be beneficial to perform the western blot and check the levels with specific antibodies, which will make the results more conclusive. However, I liked the dose-response curve of DDR genes.

We agree that protein levels by western blot could add value. It was however not possible within the time frame of this revision. It would also require more material/blood and is usually regarded as a less quantitative method, where individual differences may be more difficult to detect.

2. Micronucleus assay figure is needed other than the graph. It will be nice to see the micronuclei. The methodology of this assay is very well written, but one figure will be good to put in the main figure.

Representative MN images have now been added to this figure.

3. The authors should add an experiment to include both of the inhibitors (DNA-PK mixed with ATMi) for the DDR gene response to irradiation. This would tell whether hyperactivation of ATM upon DNA-PKcs inhibition in response to DNA damage occurs in donors 3 and 4 or not.

It would be an interesting future experiment. We have added this in the text as well as referred to a paper in the end of the discussion where dual inhibition was tested in breast cancer cells, where the net result was most similar to ATM inhibition only.

Round 2

Reviewer 1 Report

I have no furhter comments

Author Response

Thank you

Reviewer 2 Report

I am mostly satisfied with the answers of the authors to my comments.

Just one minor point: in figure 6, I thinkt hat the effects of inihbitors in absence of radiation should also be represented, if only to show that a lesser induction is not due to already a lesser expression of the gene in control conditions. If this is not the case, this point should be addressed.

Author Response

Thank you for this valid question. We have now recalculated the data in Supplemental figure 1 and related it to the average control level without inhibitor instead, to visualise the control levels with inhibitor vs without, and to compare the overall pattern for the genes.

Importantly, no change in gene expression was induced by the inhibitors only (without IR) for these genes. This is now shown in Supplemental figure 2, and it principially looks the same as Supplemental figure 1. The main difference is that the standard deviations became slightly smaller, which is an effect that sometimes is seen when controls no longer are forced to 1. An explanation was added in the text as well to refer to Supplementary figure 2.

Reviewer 3 Report

No further concerns!

Author Response

Thank you